# Reduced Consumption of Sugar-Sweetened Beverages Is Associated with Lower Body Mass Index Z-Score Gain among Chinese Schoolchildren

**DOI:** 10.3390/nu14194088

**Published:** 2022-10-01

**Authors:** Chenchen Wang, Yijia Chen, Xin Hong, Hao Xu, Hairong Zhou, Weiwei Wang, Nan Zhou, Jinkou Zhao

**Affiliations:** 1Department of Non-Communicable Chronic Disease Prevention, Nanjing Municipal Center for Disease Control and Prevention, Nanjing 210003, China; 2Department of Non-Communicable Chronic Disease Prevention, Jiangsu Provincial Center for Disease Control and Prevention, Nanjing 210009, China

**Keywords:** sugar-sweetened beverages, body mass index, body mass index z-score, schoolchildren, overweight, obesity, China

## Abstract

To examine whether reducing sugar-sweetened beverage (SSB) consumption is associated with reduced body mass index z-score gain among Chinese schoolchildren in Nanjing, China, a randomized controlled trial (RCT) was conducted in four selected primary schools from September 2019 to September 2020. Students in the third grade in the Intervention Group received school-based and home-based interventions for two consecutive semesters to reduce SSB consumption, while two schools in the Control Group did not receive any interventions. Weight changes were expressed as body mass index (BMI) z-scores as standard deviations of the BMI distribution per age and sex group. Changes in SSB consumption before and after the interventions were categorized into Level-Up if it increased, Level-Same if it was maintained and Level-Down if it decreased. Multivariable linear regression models were used to explore the association of different levels of changes in SSB consumption pre- and post-intervention with the BMI z-score. Among 1633 participants who completed the trial, the mean age at baseline was 9.36 years (±0.48 SD).The median baseline BMI z-score was −0.24 (25th percentile −0.72; 75th percentile 0.58). After the intervention, the median BMI z-score increased by 0.06 (−0.17~0.37) in the Intervention Group and by 0.14 (−0.08~0.41) in the Control Group (*p* < 0.001). A higher increase in BMI was found in the Control Group than in the Intervention Group (1.20 vs. 0.94) during the 12-month period. Among participants whose parents’ educational attainment was above 9 years, the median BMI z-score increased by 0.07 (−0.17~0.37) in the Intervention Group and by 0.16 (−0.06~0.41) in the Control Group (*p* < 0.001). In a linear regression analysis adjusted for potential confounders, the BMI z-score decreased by 0.057 more in Level-Down than in Level-Up (95% CI: −0.103 to −0.012, *p* = 0.014). These results indicate that the decreased consumption of SSBs might have reduced the prevalence of overweight in schoolchildren in China, especially in students whose parents had high educational levels.

## 1. Introduction

During the past decades, childhood overweight and obesity have become major public health concerns [1,2,3,4,5,6,7,8]. The prevalence of overweight and obesity among children and adolescents aged 5–19 has risen dramatically, from just 4% in 1975 to just over 18% in 2016, with more than 340 million children and adolescents being overweight or obese, as reported by the World Health Organization (WHO) [1]. From 1978 to 2016, the prevalence of childhood obesity more than tripled, increasing from 5 percent to 18.5 percent, inchildren aged 2–19 years in the United States [5]. Similar trends in childhood overweight and obesity were reported in European and Asian countries and the Mediterranean region [6,7,8].

In China, 19.0% of children aged 6–17 years were reportedly overweight or obese, according to the Report on Chinese Residents’ Chronic Diseases and Nutrition (2020), compared with 16.0% in 2015 [9,10]. In 2017, 17.8% of children aged 6–18 years were overweight, and 13.8% were obese, as revealed in a cross-sectional survey among 32,005 students in Jiangsu province in eastern China [11].

Overweight and obesity have adverse consequences in schoolchildren both physically, such as breathing difficulties, an increased risk of fractures, hypertension, abnormal fasting glucose, and early markers of cardiovascular diseases, and mentally, such as anxiety, tension, and loneliness [1,12,13]. The occurrence of overweight and obesity has attracted much attention because of its persistence into adulthood [3,4]. As a serious public health problem among adults, overweight and obesity are known risk factors for many non-communicable diseases [14]. 

WHO recognized that the fundamental cause of obesity and overweight is an energy imbalance between calories consumed and calories expended [1]. Children’s unhealthy weight gain is mainly associated with an increased intake of energy-dense foods that are high in fat and sugars [15,16]. Dietary interventions are required to prevent excess weight gain in childhood by limiting energy intake from foods high in sugars [17,18]. The consumption of sugar-sweetened beverages (SSBs) has been associated with weight gain and overweight in most observational studies, but not all studies [19,20]. SSBs are major sources of calories because they are less satiating than solid food, so the consumption of other foods is not reduced after consuming SSBs [21]. Possibly due to the different dietary practices in Asia, SSB drinkers consumed more total energy and showed a higher prevalence of obesity in Korean boys 7–12 years of age and Chinese children [22,23]. 

In recent years, SSB consumption among children has risen globally [8,9,10,17,23].The China Nutrition and Health Survey (2010–2012) showed that 61.9% of children aged 6–17 years old drank SSBs at least once a week [10]. Over the past two decades, more public health interventions have been implemented to reduce SSB consumption in children [24,25,26,27,28,29]. There is an ongoing debate around whether there is sufficient scientific evidence that the decreased consumption of SSBs would reduce obesity. A randomized trial by Ebbeling et al. noted that when replacing SSBs with noncaloric beverages, the increase in BMI was smaller in the experimental group at the end of the first-year intervention period, but not for the 2-year period [26]. The study by Sichieri et al. focused on reducing the consumption of sugar-sweetened carbonated beverages by students aged 9–12 years and found that BMI was only significantly reduced among overweight girls [27]. Cunha et al. found that encouraging the adoption of healthy eating led to a reduction in SSBs but did not lead to a reduction in BMI gains in Brazil [28].

In China, most studies examining the relationship between SSB consumption and body weight are cross-sectional by design [15,30,31]. Despite studies in Brazil and other countries, there is still inadequate evidence on whether reducing the consumption of SSBs can prevent overweight and obesity among Chinese students to inform possible policy changes in China. The main purpose of this study was to test the hypothesis that reducing SSB consumption is associated with reduced body mass index z-score gain among schoolchildren in Nanjing, the capital of Jiangsu province, China.

## 2. Materials and Methods

### 2.1. Study Subject Selection and Interventions

A randomized controlled trial (RCT) was conducted among Chinese schoolchildren in four selected primary schools from September 2019 to September 2020 in Nanjing, eastern China. The methods were described earlier in detail [32]. In summary, two schools in either rural or urban strata were randomly assigned to the Intervention Group or the Control Group. Students in the third grade of two schools in the Intervention Group received interventions for two consecutive semesters to reduce SSB consumption, while two schools in the Control Group only received regular monitoring, as in the Intervention Group, but without any interventions.

Students spend most of their weekdays in school, while parents are both role models and key decision-makers for children’s food choices. The intervention measures were therefore combined school-based and home-based interventions to reduce SSB consumption. School-based interventions included health education courses, school environment support, class environment support and a sugar-free school campaign. Home-based interventions included health lectures, instant core messages, little hands holding big hands and student–parent collaboration [32].

### 2.2. Data Collection and Outcome Measures

The same questionnaire was self-administrated by selected students at baseline (September 2019) and at the end of the intervention (September 2020), facilitated by a standard PowerPoint slide set in each classroom. The study team, comprising local CDC staff, a health care teacher and a class teacher, was trained by the principal investigator. The questionnaire included demographic characteristics (date of birth, sex, the name of the school, school grade and school class), parental information (parental education level, height and weight), student activity level (weekly accumulated physical activity outside school, daily cumulative homework time, screen time and sleep time), weekly frequency of SSB consumption and average intake each time. The weekly frequency of SSB consumption and average intake each time was based on the Food Frequency Questionnaire (FFQ) from the China Nutrition and Chronic Disease Health Survey, which was used in a previous study in China [31]. We conducted a validity test: Cronbach’s Alpha of parental information = 0.759; Cronbach’s Alpha of student activity behavior = 0.69; Cronbach’s Alpha of SSBs consumption = 0.706; Kaiser–Meyer–Olkin = 0.88; and P Bartlett < 0.001. 

Parental information (education level, weight and height) was self-reportedwhen obtaining their informed consent. Children’s weights and heightswere measured by trained nurses at the Community Healthcare Centers at baseline and at the end of the intervention using electronic body scales and mechanical anthropometry stadiometers (Bengbu Equipment; Bengbu, China). Both instruments were calibrated on a regular basis according to the standard protocol, and measurements were to the nearest 0.1 cm or kg, respectively.

### 2.3. Categorization of Children into Different Levels of Changes in SSB Consumption

Broad beverage categories were used, categorized based on nutrient content and on China’s Beverage General Rule (GB10789-2008). The consumption of SSBs was measured on a seven-day frequency scale, using the question ‘How many times and how much on average each time did you drink carbonated beverages in past week, commonly available in the market?’. The amount of SSB consumption was calculated when any of the 8 categories of SSB consumption was reported. The total amount of SSB consumption was calculated by multiplying the frequency of intake by the average amount consumed each time. According to changes in SSB consumption before and after the SSB intervention, the consumption of SSBs was categorized into the Level-Up Group if it increased after the intervention, the Level-Same Group if it remained unchanged and the Level-Down Group if it decreased.

### 2.4. Data Management and Statistical Analysis

Self-reported parents’ and measured children’s heights and weights were used to calculate body mass index (BMI) using the formula: BMI = weight (kg)/height (m)^2^. Because the standard deviation (SD) of children’s BMI increases with age, children’s BMIs were converted to BMI z-scores, derived from the Chinese National Survey on Students Constitution and Health in 2014, with both age and sex accounted for [33].

All statistical analyses were performed using the statistical software package IBM SPSS Statistics Version 20.0 (SPSS Inc., Chicago, IL, USA). Qualitative variables are described in absolute frequencies and percentages. Quantitative variables conforming to a normal distribution are described as means and SDs. Medians, together with the 25th and 75th percentiles, were used for those not conforming to a normal distribution. Pearson’s chi-square test was used to compare categorical variables. The independent t-test was used to compare normally distributed continuous variables, while the Mann–Whitney test was used to compare abnormally distributed continuous variables. Multivariable linear regression models were used to explore the association of changes in SSB consumption during the intervention period with the BMI z-score. In the analyses, potential confounders were added to the crude model (Model 1) in three steps. Model 2 was adjusted for child factors (age and gender); Model 3 was further adjusted for parental lifestyle factors (area, parental education level, father’s BMI and mother’s BMI; Model 4 was further adjusted for differences in children’s individual behaviors, including physical activity time outside school, homework time, screen time and sleep time. A *p* value below 0.05 was considered statistically significant.

## 3. Results

### 3.1. Demographic and Clinical Characteristics

At baseline, out of 1633 participants, 874 (53.58%) were boys and 758 (46.42%) were girls, with a mean age of 9.36 years (±0.48 SD). The percentage of participants living in rural areas (n = 891, 54.56%) was slightly higher than that of those living in urban areas (n = 742, 45.44%). Two-thirds (66.81%) of parents had more than nine years of education. At baseline, the mean height, weight and BMI of students were 134.15 (±6.60 SD), 31.19 (±7.25 SD) and 17.16 (±2.93 SD), respectively. The median baseline BMI z-score was −0.24 (25th percentile −0.72; 75th percentile 0.58). There were no significant differences in mean age, sex, parents’ educational attainment, parents’ BMI, physical activity time outside school, homework time, screen time, anthropometric measures (height and weight), BMI z-score or SSB consumption at baseline between students in the two groups. Students in the Intervention Group were more likely to spend more time sleeping than those in the Control Group (Table 1).

### 3.2. Changes in BMI z-Score

Table 2 describes the changes in the BMI z-score, height, weight and BMI between the Intervention and Control Groups among 1633 primary school students from baseline to the end of the study. The median BMI z-score increased by 0.06 (−0.17~0.37) in the Intervention Group and by 0.14 (−0.08~0.41) in the Control Group (*p* < 0.001). 

The median BMI z-score increased by 0.07 (−0.17~0.37) in the Intervention Group and by 0.16 (−0.06~0.41) in the Control Group (*p* < 0.001) when parents’ educational attainment was above 9 years. The median BMI z-score increased by 0.05 (−0.18~0.35) and by 0.11 (−0.11~0.41) in the Intervention Group and the Control Group, respectively, when parents’ educational attainment was ≤9 years or below, with no significant difference (*p* = 0.103).

### 3.3. Changes in BMI Z-Score with Different Levels of Changes in SSB Consumption

Table 3 presents the differences in changes in BMI z-scores before and after the intervention between the Control Group and Intervention Group, with different levels of changes in SSB consumption. In Level-Down, the median BMI z-score increased by 0.03 (−0.18~0.33) in the Intervention Group and by 0.14 (−0.08~0.40) in the Control Group (*p* = 0.001). In Level-Same, the median BMI z-score increased by 0.04 (−0.24~0.30) in the Intervention Group and by 0.17 (−0.10~0.52) in the Control Group (*p* = 0.044), while in Level-Up, the median BMI z-score increased by 0.10 (−0.15~0.41) in the Intervention Group and by 0.15 (−0.07~0.42) in the Control Group (*p* = 0.178). When comparing the differences in changes in BMI z-scores for different levels of changes in SSB consumption before and after the intervention, the median BMI z-score increased by 0.09 (−0.14~0.37) in Level-Down and by 0.13 (−0.10~0.41) in Level-Up, with a median difference of 0.04 (*p* = 0.049).

### 3.4. Differences in Changes in BMI Z-Score with Different Behaviors of SSB Intake

In a linear regression analysis adjusted for potential confounders, the BMI z-score among 1633 primary school students decreased by 0.057 more in Level-Down than in Level-Up (95% CI: −0.103 to −0.012, *p* = 0.014) (Table 4).

## 4. Discussion

Our study found that the reduced consumption of SSBs is associated with lower BMI z-score gain in Chinese schoolchildren aged 9–10 years. To our best knowledge, this is the first trial to examine the effects of reducing the consumption of SSBs on BMI z-score gain among Chinese schoolchildren [3,15,17,30,31,34].

BMI, calculated from weight and height, is often used as a surrogate for adiposity. As schoolchildren undergosignificant changes in height and weight for physiological reasons, the BMI z-score, instead of absolute BMI, was used as an indirect age- and sex-specific measure. A positive change in the BMI z-score during the trial indicates an increase in relative BMI over the time interval. In our study, the Control Group had a significantly greater increase in the median BMI z-score (0.14) as compared to that in the Intervention Group (0.06), although BMI z-scores increased in both groups. The level of the change in SSB consumption before and after the intervention was associated with the change in the BMI z-score. In the Level-Down Group, the BMI z-score decreased by 0.057 more than that in the Level-Up Group. 

The reduction in children’s BMI z-score gain in the present study was associated with the parents’ educational level. The higher the parental educational levels, the lower the increase in children’s BMI z-scores in the Intervention Group. Parental involvement has been widely recognized as a vital strategy for improving children’s physical health, behavior and mental development in general [35,36]. Most of the main meals for Chinese students are in the evening with their parents. This emphasizes that home-based intervention may play an important role in achieving a reduction in excess weight gain or overweight development in schoolchildren [37,38,39,40]. In our study, parents’ knowledge was increased through health lectures for parents and regular instant messages through WeChat or QQ in home-based interventions. The family beverage-drinking environment was improved in the Intervention Group, such as parents restricting their children from drinking SSBs and reducing the stock of SSBs at home [32]. Previous intervention studies have indicated that educating parents could potentially increase parental health behaviors and improve parental efficacy, with especially greater effects among parents with higher education levels [38,41,42]. This was further supported by a study revealing that parents who had higher levels of risk perception and parental efficacy were more likely to positively improve children’s health behaviors [37]. The findings suggest that differentiated interventions for parents with different levels of education could be considered in the future [42]. 

Our data showed that the difference in weight gain was minimal and insignificant, with an approximately 0.13 kg difference in weight gain between the two study groups. Students in the Intervention Group grew approximately 0.97 cm taller than those in the Control Group. Therefore, this led to a more pronounced reduction in BMI z-score gain in Intervention Group children. This is not entirely consistent with previous studies. Previous studies by Ruyter et al. and Ebbeling et al. reported that after an intervention designed to decrease the consumption of SSBs, the intervention group had significant reductions in weight gain (−1.01 kg and −1.9 kg, respectively) compared to the control group [24,26]. However, the difference in height gain was minimal between the two groups. SSBs not only contain high amounts of added sugars but also are associated with unhealthy food patterns or eating behaviors [43]. A previous study by Keller et al. suggested that increased SSB intake is associated with reduced milk and calcium intake [44]. Lee et al. reported that SSB intake was inversely associated with milk, fruit and vegetable intake in Korean children and adolescents [22]. It may be arguably true that the Intervention Group’s limitation of SSB consumption may have helped increase milk and dairy consumption, which is positively associated with height in adolescents [45,46]. This suggests that further research should examine the effect of a longer intervention or a delayed post-intervention assessment of reductions in SSB consumption on student height.

After adjusting for potential confounders, the level of changes in SSB consumption pre- and post-intervention was clearly associated with changes in the BMI z-score. In the Level-Down Group, the BMI z-score decreased by 0.057 more than that in the Level-Up Group. The findings suggest that the smaller increase in the BMI z-score might be associated with the reduced consumption of SSBs. Our findings are consistent with recent reviews reporting that reducing the consumption of SSBs reduced weight gain and obesity in schoolchildren [18,24,25]. The excess intake of free sugars (all sugars added to foods) is a key contributor to unhealthy weight gain [17]. Ruyter et al. indicated that the reduced ingestion of liquid sugars might also reduce the insulin spike and thus diminish hunger [24]. The removal of liquid sugar was not sensed by satiating mechanisms and was incompletely compensated for by the increased consumption of other foods [24,25,34]. Therefore, reducing SSB sugar intake could prevent excess weight gain in childhood by limiting excess energy intake.

Our findings are consistent with previous research by Ruyter et al. and Katan et al. in an 18-month double-blind, randomized controlled trial involving 641 primary school children from 4 to 11 years of age in the United States. The replacement of one 250 mL drink per day with a sugar-free beverage significantly reduced weight gain and fat accumulation, which may benefit a large proportion of children [24,25].However, some analyses of RCTs conducted encouraging the adoption of healthy eating promoted a reduction in SSBs but did not lead to a BMI reduction. A previous study by Cunha et al. delivered nine nutritional education sessions by trained nutritionists in Brazil. The 9-month intervention led to significant variation in the daily frequency of consumption of SSBs among students aged 11 years but did not result in a reduction in BMI gain. However, the two groups were not well matched at baseline, with a higher prevalence of being overweight and obese and a slightly higher intake of SSBs among participants in the control group [28]. Previous studies identified that the intake of free sugars or SSBs is a determinant of body weight. When considering the rapid weight gain that occurs after an increased intake of sugars, it seems reasonable to conclude that advice relating to sugar intake is a relevant component of a strategy to reduce the high risk of overweight and obesity in most countries [47,48].

Our study has several limitations. First, this study selected four primary schools, which may not represent all primary schools in Nanjing. Second, we did not collect the total food intake of the students and could not assess actual dietary intakes of total energy and food components. It is possible that associations between SSBs and BMI were partly confounded by other aspects of the diet, such as the intake of other categories of energy-dense foods. Furthermore, we could not assess the relationship between SSB consumption and dairy consumption [49]. Third, we collected physical activity time outside of the school but not inside the school. This makes it impossible to assess total physical activity time as a confounder. Fourth, the long-term effects of the intervention could not be assessed. Fifth, parental body weight and height were self-reported, which is prone to reporting and/or recall biases. Finally, we did not statistically adjust for clustering, which may lead to an inevitable potential for bias in allocation at the school level. Despite all of the above-mentioned limitations, this study included a large sample size and achieved a high follow-up rate over a period of 12 months. The use of the BMI z-score has been considered preferable to the use of absolute BMI [50], and many studies reported their results in terms of changes in z-scores [49,51]. The trial provides important learning for consideration in the design of future weight control interventions and provides important information for policy makers and educators responsible for promoting healthy weight in Chinese students.

## 5. Conclusions

In conclusion, the results of our study show that the decreased consumption of SSBs might be associated with the reduced prevalence of overweight in schoolchildren in China, especially in students whose parents have high educational levels. The design for combined school-based and home-based interventions for reducing SSB consumption may be applicable for overweight interventions in schoolchildren in the future. To achieve long-term benefits, the intervention needs to be translated into sustained healthy behaviors over time. Future studies should examine whether a longer intervention or delayed post-intervention assessment would lead to the same level of reduction in body weight gain among schoolchildren.

## Figures and Tables

**Table 1 nutrients-14-04088-t001:** Demographic characteristics of subjects at baseline.

Characteristics	Total	Intervention Group (n = 887)	Control Group (n = 746)	*p* Value
Sex				
Male(n/%)	875/53.58	480/54.11	395/52.95	*χ*^2^ = 0.221, *p* = 0.654 ^e^
Female(n/%)	758/46.42	407/45.89	351/47.05	
Age (years)	9.36 ± 0.48	9.34 ± 0.48	9.38 ± 0.49	*t* = −1.752, *p* = 0.080 ^f^
Area				
Rural(n/%)	891/54.56	516/58.17	375/50.27	*χ*^2^ = 10.215, *p* = 0.001 ^e^
Urban(n/%)	742/45.44	371/41.83	371/49.73	
Parental education level				
≤9 years(n/%)	542/33.19	298/33.60	244/32.71	*χ*^2^ = 0.895, *p* = 0.629 ^e^
>9 years(n/%)	1091/66.81	589/66.40	502/67.29	
Father’s BMI ^b^ (kg/cm^2^) ^a^	24.65 ± 3.36	24.76 ± 3.61	24.51 ± 3.42	*t* = 1.470, *p* = 0.142 ^f^
Mother’s BMI (kg/cm^2^) ^a^	21.83 ± 3.21	21.92 ± 3.42	21.72 ± 2.93	*t* = 1.280, *p* = 0.142 ^f^
Physical activity time outside school (minutes/week) ^a^	217.33 ± 221.92	221.74 ± 229.42	212.07 ± 212.66	*t* = 0.876, *p* = 0.381 ^f^
Homework time(minutes/day) ^a^	116.26 ± 63.15	115.14 ± 64.67	117.60 ± 61.31	*t* = −0.785, *p* = 0.432 ^f^
Screen time (minutes/day) ^a^	30.19 ± 50.69	28.43 ± 45.44	32.30 ± 56.28	*t* = −1.533, *p* = 0.126 ^f^
Sleep time(hours/day) ^a^	9.15 ± 1.07	9.23 ± 1.02	9.06 ± 1.13	*t* = 3.259, *p* = 0.001 ^f^
High (cm) ^a^	134.15 ± 6.60	133.96 ± 6.56	134.37 ± 6.66	*t* = −1.236, *p* = 0.217 ^f^
Weight (kg) ^a^	31.19 ± 7.25	30.98 ± 7.33	31.43 ± 7.16	*t* = −1.224, *p* = 0.221 ^f^
BMI (kg/m^2^) ^a^	17.16 ± 2.93	17.11 ± 2.96	17.23 ± 2.89	*t* = −0.851, *p* = 0.395 ^f^
BMI z-score ^c^	−0.24(−0.72,0.58)	−0.24(−0.75,0.57)	−0.22(−0.67,0.59)	*z* = −1.001, *p* = 0.317 ^g^
SSB ^d^ consumption (ml/week) ^c^	750.00 (250.00,1750.00)	750.00 (250.00,1750.00)	750.00 (250.00,1750.00)	*z* = −1.276, *p* = 0.202 ^g^

^a^ Mean ± SD; ^b^ BMI = body mass index; ^c^ median (25th percentile; 75th percentile); ^d^ SSBs = sugar-sweetened beverages; ^e^ chi-square test; ^f^
*t*-test; ^g^ Mann–Whitney test.

**Table 2 nutrients-14-04088-t002:** Changes in BMI z-score, height, weight and BMI between two groups among 1633 primary school students during the study in Nanjing City, China.

Outcome	Intervention Group (n = 887)			Control Group (n = 746)			Difference in Change from Baseline (95% CI)	*p* Value for Difference
	0 mo	12 mo	Change	0 mo	12 mo	Change		
All children who completed study (n = 1633)								
BMI ^a^ z-score ^c^	−0.24 (−0.75~0.57)	−0.10 (−0.70~0.76)	0.06 (−0.17~0.37)	−0.22 (−0.67~0.59)	0.02 (−0.64~0.86)	0.14 (−0.08~0.41)	−0.08 (−0.12~−0.04)	*z* = −4.130, *p* < 0.001
Height (cm) ^b^	133.97 ± 6.51	141.76 ± 6.44	7.79 ± 3.62	134.47 ± 6.58	141.30 ± 6.64	6.82 ± 2.19	0.97 (0.67~1.26)	*t* = 6.365, *p* < 0.001
Weight (kg) ^b^	30.98 ± 7.33	36.56 ± 8.68	5.58 ± 3.06	31.41 ± 7.14	37.12 ± 8.89	5.71 ± 2.90	−0.13 (−0.4 3~0.16)	*t* = −0.913, *p* = 0.361
BMI (kg/m^2^) ^b^	17.11 ± 2.96	18.05 ± 3.34	0.94 ± 1.40	17.23 ± 2.89	18.42 ± 3.34	1.20 ± 1.27	−0.26 (−0.38~−0.12)	*t*= −3.815, *p* < 0.001
Parental education level ≤ 9 years(n = 542)								
BMI z-score ^c^	−0.33 (−0.82~0.52)	−0.21 (−0.73~0.70)	0.05 (−0.18~0.35)	−0.27 (−0.61~0.60)	0.04 (−0.62~0.85)	0.11 (−0.11~0.41)	−0.06 (−0.14~−0.01)	*z* = −1.629, *p* = 0.103
Height (cm) ^b^	132.95 ± 6.30	141.22 ± 6.06	8.27 ± 3.69	133.08 ± 6.09	140.20 ± 6.32	7.13 ± 2.43	1.14 (0.60~1.68)	*t* = 4.142, *p* < 0.001
Weight (kg) ^b^	30.24 ± 7.40	35.86 ± 8.56	5.62 ± 2.91	30.73 ± 6.71	36.38 ± 8.69	5.64 ± 3.17	−0.02 (−0.54~0.49)	*t* = −0.091, *p* = 0.928
BMI (kg/m^2^) ^b^	16.94 ± 3.03	17.84 ± 3.30	0.89 ± 1.34	17.23 ± 2.86	18.35 ± 3.41	1.12 ± 1.40	−0.23 (−0.46~0.00)	*t* = −1.942, *p* = 0.053
Parental education level > 9 years(n = 1091)								
BMI z-score ^c^	−0.18 (−0.72~0.66)	−0.04 (−0.69~0.82)	0.07 (−0.17~0.37)	−0.18 (−0.71~0.57)	0.06 (−0.65~0.86)	0.16 (−0.06~0.41)	−0.09 (−0.14~−0.03)	*z* = −3.892, *p* < 0.001
Height (cm) ^b^	134.49 ± 6.56	142.04 ± 6.61	7.55 ± 3.56	135.15 ± 6.70	141.83 ± 6.73	6.68 ± 2.05	0.87 (0.52~1.22)	*t* = 4.834, *p* < 0.001
Weight (kg) ^b^	31.36 ± 7.27	36.92 ± 8.73	5.55 ± 3.13	31.74 ± 7.33	37.49 ± 8.97	5.74 ± 2.76	−0.19 (−0.54~0.16)	*t* = −1.058, *p* = 0.290
BMI (kg/m^2^) ^b^	17.19 ± 2.93	18.15 ± 3.36	0.97 ± 1.42	17.23 ± 2.91	18.46 ± 3.31	1.23 ± 1.20	−0.26 (−0.42~−0.11)	*t* = −3.289, *p* = 0.001

^a^ BMI = body mass index; ^b^ mean ± SD; ^c^ median (25th percentile, 75th percentile).

**Table 3 nutrients-14-04088-t003:** Differences in changes inBMI z-scores before and after intervention between the Control Group and the Intervention Group with different levels of changes in SSB consumption.

Behavior of SSB Consumption	Number (n/%)	BMI Z-Score ^a^			t	*p* Value for Difference
		Before the Intervention	After the Intervention	Change ^b^		
Level-Up ^c^						
Control Group	346/55.90	−0.30 (−0.70,0.59)	−0.06 (−0.65,0.86)	0.15 (−0.07,0.42)	−1.348	0.178
Intervention Group	273/44.10	−0.37 (−0.83,0.48)	−0.19 (−0.72,0.70)	0.10 (−0.15,0.41)		
Level-Same ^d^						
Control Group	56/30.94	−0.15 (−0.65,0.69)	0.16 (−0.61,0.79)	0.17 (−0.10,0.52)	−2.016	0.044
Intervention Group	125/69.06	−0.21 (−0.80,0.75)	−0.12 (−0.69,0.90)	0.04 (−0.24,0.30)		
Level-Down ^e^						
Control Group	344/41.30	−0.15 (−0.64, 0.56)	−0.05 (−0.63,0.86)	0.14 (−0.08, 0.40)	−3.217	0.001
Intervention Group	489/58.70	−0.19 (−0.67,0.62)	−0.04 (−0.68,0.76)	0.03 (−0.18,0.33)		
Total						
Level-Up	619/37.91	−0.34 (−0.78, 0.53)	−0.12 (−0.69,0.81)	0.13 (−0.10,0.41)	6.021	0.049
Level-Same	181/11.08	−0.18 (−0.78,0.74)	−0.02 (−0.66,0.82)	0.06 (−0.17,0.38)		
Level-Down	833/51.01	−0.17 (−0.67,0.57)	0.01 (−0.67,0.80)	0.09 (−0.14,0.37)		

^a^ Median (25th percentile, 75th percentile); ^b^ change = change in BMI z-score, after intervention–before intervention; according to changes in SSB consumption before and after the SSB intervention; ^c^ Level-Up: the consumption of SSBs increased after the intervention; ^d^ Level-Same: the consumption of SSBs was unchanged; ^e^ Level-Down: the consumption of SSBs wasreduced.

**Table 4 nutrients-14-04088-t004:** Linear regression model of the association of difference in changes in BMI z-score with different behaviors of SSB intake (n = 1633).

Model	Levels of Changes in SSB Consumption	*β*	95% CI	*p* Value
1 ^a^	Level-Up	1		
	Level-Same	−0.043	−0.115, 0.028	0.236
	Level-Down	−0.046	−0.091, −0.001	0.044
2 ^b^	Level-Up			
	Level-Same	−0.036	−0.108, 0.035	0.381
	Level-Down	−0.048	−0.092, −0.003	0.037
3 ^c^	Level-Up			
	Level-Same	−0.027	−0.100, 0.045	0.461
	Level-Down	−0.055	−0.100, −0.010	0.018
4 ^d^	Level-Up			
	Level-Same	−0.028	−0.100, 0.045	0.481
	Level-Down	−0.057	−0.103, −0.012	0.014

Values of β and 95% confidence intervals (CIs) are from linear regression analyses, with the levels of changes in SSB consumption in the Level-Up Group as the reference. ^a^ Model 1: Level of SSB consumption; ^b^ Model 2: Model 1 + age and gender; ^c^ Model 3: Model 2 + area, parental education level, father’s BMI and mother’s BMI; ^d^ Model 4: Model 3 + physical activity time outside school, homework time, screen time and sleep time.

## Data Availability

The data presented in this study are available from the corresponding authors on reasonable request.

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
