# Peer review of "Reduced Consumption of Sugar-Sweetened Beverages Is Associated with Lower Body Mass Index Z-Score Gain among Chinese Schoolchildren"

_nutrients, 2022, doi:10.3390/nu14194088_

Round 1

Reviewer 1 Report

The author present a manuscript about an interesting research focused on "Sugar-sweetened beberages consumption in children"

My first to commentaries are about ethical statements. There is absence of information about the application of the items of SPIRIT 2013 Statement. It must be defined in the manuscript.

As this is an interventional research, it must be registered in an international database repository such as clinicaltrials.gov. There is no information about it.

TITLE

Using "appears" in title sounds like if you didn't use any statistical analysis.

ABSTRACT

You don't define briefly what does the RCT consisted in.

INTRODUCTION
The sentences of first paragraph are extremely short, I consider that you repeat information between them, it is confuse.

Line 66: Low in nutrients? Do you consider that fat and sugar are not nutrients?

Line 79: Last "two" decades

SUBJECTS AND METHODS

Is it appropiate to put "Subjects" in the section title?

I understand that you have published the RCT before, but you must describe again the principal RCT points in this new manuscript.

Line 99: "Two" schools

Lines 111-113: Why didn't you measure parents body weight and height? I consuder that this must represent a very important limitation to the study.

Is the questionnaire validated before?

RESULTS

Table 1: (n/%) is not appropiated in the title. What does "x2/t" and "P" means? BMI in kg/cm2? High?

Lines 174-178: I consider not necessary to mention height ans weight changes (BMI z score is enough).

Table 2: HIgh? I consider that BMI z score is the only important information in this Table, in 9-10 years old children there will be significative changes in height and weight for physiological reasons, or you must add discussion about the importance of report changes in those parameters.

Again, even you have a previous manuscript where you mention the interventions, it is very necessary to describe them again, including the importance of the parental educational level.

Table 3. Number of what? Bold use is confusing.

DISCUSSION

I don't see in the discussion section any information to stablish the possible importance of Table 4.

Line 233: "et" al.

There appears that weight gain in children is unhealthy, but weight gain is very necessary for good health in children, you must focus some discussion about it and the BMI z score results.

Why you are sure that the effects of the interventions are related to SSBs decrease and not the eating pattenrs? Those were not considered in Table 4 as potential confounders.

I already se that my last comment is mentioned as a limitation, but more discussion about it ss necessary.

Line 289: Use abbreviation "SSBs"

You mention that the lack of physical activity information is a limitation, but you used ohysical activity outside school as a potential confounder in Table 4, this is cofsuse to me.

CONCLUSION/GENERAL COMMENTS

Considering that the BMI z score was calculated according the Chinese National Survey on Students Constitution and Health 2014, I understand that most of the children were in normal weight (mean under 0) at the begining and at the final of the interventions, for that reason I consider that the study should have used a sample of overweight children, due that the effects of the intervention in that population really could have clinically important results, ot you must discuss why using a sample of normal weight children is more important than one of overweight.

Author Response

Dear Editor:

Thank you for your and reviewer’s comments for our manuscript entitled ‘Reduced consumption of sugar-sweetened beverages appears to be associated with lower body mass index z-score gain among Chinese schoolchildren (No.: nutrients-1913929)’. Those comments are very helpful for revising and improving our manuscript. We revised the manuscript with tracked changes and point-to-point responses to each of comments below.

We hope these changes and responses are sufficient. Again, we appreciate the editor and reviewer’s time and efforts in reviewing this manuscript. We feel honored to be able to publish our results in Nutrients.  

 Yours sincerely,

Nan Zhou

Reviewer 2 Report

This study examines whether the reducing sugar-sweetened beverages (SSBs) consumption

is as- 15 sociated with reduced weight gain among Chinese schoolchildren. My main concern

is about the novelty of the study. As the authors already indicated a couple of times in the

discussion, their results are in line with previous studies. I suggest rewriting the rationale of

the study in the introduction and showing what this study will add to the literature or what

is new in this study.

Some minor comments are below:

Please re write the first sentence of the abstract.

Add the mean age of children into the abstract.

The authors may consider changing name Level-Even to Level-Same.

The first sentence and third sentence of the introduction are almost the same. The first one

can be removed

There is no need to give the prevalence information about other countries. I suggest

focusing on only the worldwide and China statistics.

What is the novelty of this paper except being the first longitudinal study in China? Authors

summarised other related studies in Brazil and other countries but did not mention what

this study will bring to the literature

Is it possible to say place group to the intervention group as there is no intervention? The

methods section is a bit confusing; I suggest writing it in detail, especialiy control and

intervention section.

Please add a reference for Chinese National Survey on Students Constitution and Health in

2014.

Author Response

(The authors gave the same response as above.)

Round 2

Reviewer 1 Report

I recomend to accept this revised manuscript

Reviewer 2 Report

I would like to thank the authors for addressing my comments.